# XENet: Using a new graph convolution to accelerate the timeline for protein design on quantum computers

**Jack B. Maguire** [1]*, **Daniele Grattarola** [2], **Vikram Khipple Mulligan** [3],
**Eugene Klyshko** [1,4], **Hans Melo** [1]

**1** Menten AI, Inc., Palo Alto, California, United States of America, **2** Faculty of Informatics, Università della Svizzera italiana, Lugano, Switzerland, **3** Center for Computational Biology, Flatiron Institute, New York, New York, United States of America, **4** Department of Physics, University of Toronto, Toronto, Ontario, Canada

* jbmaguire@menten.ai

**Data Availability Statement:** All relevant data are within the manuscript and its Supporting information files. The Rosetta source code itself is not publicly available however it is free for

## Abstract

Graph representations are traditionally used to represent protein structures in sequence design protocols in which the protein backbone conformation is known. This infrequently extends to machine learning projects: existing graph convolution algorithms have shortcomings when representing protein environments. One reason for this is the lack of emphasis on edge attributes during massage-passing operations. Another reason is the traditionally shallow nature of graph neural network architectures. Here we introduce an improved message-passing operation that is better equipped to model local kinematics problems such as protein design. Our approach, XENet, pays special attention to both incoming and outgoing edge attributes. We compare XENet against existing graph convolutions in an attempt to decrease rotamer sample counts in Rosetta's rotamer substitution protocol, used for protein side-chain optimization and sequence design. This use case is motivating because it both reduces the size of the search space for classical side-chain optimization algorithms, and allows larger protein design problems to be solved with quantum algorithms on near-term quantum computers with limited qubit counts. XENet outperformed competing models while also displaying a greater tolerance for deeper architectures. We found that XENet was able to decrease rotamer counts by 40% without loss in quality. This decreased the memory consumption for classical pre-computation of rotamer energies in our use case by more than a factor of 3, the qubit consumption for an existing sequence design quantum algorithm by 40%, and the size of the solution space by a factor of 165. Additionally, XENet displayed an ability to handle deeper architectures than competing convolutions.

## Author summary

Graph data structures are ubiquitous in the field of protein design, and are at the core of the recent advances in machine learning brought forth by graph neural networks (GNNs). GNNs have led to some impressive results in modeling protein interactions, but are still not as common as other tensor representations.

academic use: https://www.rosettacommons.org/software/license-and-download. Raw training data is publicly available at https://menten-ai-public.s3.us-east-2.amazonaws.com/Maguire-XENet-2021/all_training_data.tar.gz. Raw testing data is publicly available at https://menten-ai-public.s3.us-east-2.amazonaws.com/Maguire-XENet-2021/all_testing_data.tar.gz. Our best ECC model (used for quantum benchmark) is available in Keras h5 format at https://menten-ai-public.s3.us-east-2.amazonaws.com/Maguire-XENet-2021/best_ECC.h5. Our best CrystalConv model (used for quantum benchmark) is available in Keras h5 format at https://menten-ai-public.s3.us-east-2.amazonaws.com/Maguire-XENet-2021/best_CrystalConv.h5. Our best XENet model (used for quantum and classical benchmarks) is available in Keras h5 format at https://menten-ai-public.s3.us-east-2.amazonaws.com/Maguire-XENet-2021/best_XENet.h5.

**Funding:** VKM is funded by the Simons Foundation: https://www.simonsfoundation.org. The funders had no role in study design, data collection and analysis, decision to publish, or preparation of the manuscript.

**Competing interests:** I have read the journal's policy and the authors of this manuscript have the following competing interests: VKM and HM are cofounders and shareholders in Menten AI, Inc. JBM is employed by Menten AI with granted stock options. The content of this manuscript is relevant to work performed at Menten AI.

Most GNN architectures tend to put minimal emphasis on information stored on edges; however, protein modeling tools often use edges to represent vital geometric relationships about residue pair interactions. We show that a more advanced processing of edge attributes can lead to considerable benefits when modeling chemical data.

We introduce XENet and show it to have improved ability to represent protein structural data while allowing information about amino acid interactions to be stored on graph edges. We use XENet to intelligently simplify the optimization problem that is solved when designing proteins. This task is important to us and others because it allows larger proteins to be designed on near-term quantum computers. We show that XENet is able to train on our protein modeling data better than existing methods, successfully resulting in a dramatic decrease in protein design sample space with negligible loss in quality.

## Introduction

Protein design involves astronomically large search problems beyond the capabilities of even the largest supercomputers. [1] This task traditionally involves assuming a static protein backbone and representing all candidate side-chain conformations and identities as discrete possibilities called "rotamers". [2–4] A single sequence position on the protein can have hundreds of candidate rotamers when spanning all twenty native amino acids. The challenge is to find one rotamer per variable position such that the value of some scoring function (typically an approximation of the conformational energy of the structure given the rotamer selection) is minimized. If the scoring function can be expressed as a sum of single-rotamer scores and two-rotamer interaction scores, much of the expense of calculating this function can be shifted to a pre-calculation which all possible one- and two-body energies are computed and stored in a lookup table.

A selection of one rotamer per position that optimizes the scoring function may be found by one of a number of approaches. For trivially small rotamer optimization problems, exhaustive enumeration is feasible, but grows infeasible for most real-world problems since the number of possible solutions given $N$ variable amino acid positions and $D$ rotamers per position is $D^N$, scaling exponentially. Exact algorithms with strong guarantees of convergence to the global optimum, such as those implemented in the Toulbar2 cost function solver, represent an alternative that makes somewhat larger problems tractible. [5] Some of these algorithms, including the dead end elimination/A* approach implemented in the Osprey protein design package, can be efficiently parallelized on CPUs or GPUs, offering an up to $M$-fold speedup given $M$ computing cores. [6] Nevertheless, these algorithms are also limited by the exponential scaling of the solution space, which rapidly outpaces any speedup from parallelism. For problems with tens of variable positions and thousands to tens of thousands of total rotamers, it is necessary to use heuristic methods that do not offer guarantees of finding the global optimum, such as the simulated annealing approaches implemented in the Rosetta software suite. [7, 8] Because a protein designer is often interested in diverse near-optimal solutions rather than in the single unique solution that optimizes the scoring function (which itself is often approximate, so that the global optimum may not be the very best solution in reality), protein designers often sacrifice the guarantee of finding the global optimum in favor of having a convenient means of rapidly sampling from the pool of near-optimal solutions. Indeed, recent work has also sought means of sampling diverse near-optimal solutions, rather than a single globally optimal solution, with exact solvers like Toulbar2 [9]. Given the exponentially-scaling solution space, however, even heuristic methods cease to be effective at sampling the low-

energy solutions for rotamer optimization problems with hundreds of variable positions or hundreds of thousands of total rotamers.

Quantum computing offers a new alternative for solving these complex combinatorial problems to power the development of new protein-based therapeutics and enzymes of industrial interest—one with the prospect of scaling efficiently to much larger design problems. [10] A register of $Q$ qubits can, on measurement, adopt one of $2^Q$ strings of 1s and 0s. Prior to measurement, it can exist in a superposition of all of these $2^Q$ possible observable states. If solution states to a rotamer optimization problem can be mapped to qubit states, then a quantum computer with a minimum of $N \log_2 D$ qubits could in theory simultaneously consider all $D^N$ solutions for a rotamer optimization problem with $N$ variable amino acid positions and $D$ rotamers per position. A suitable quantum optimization algorithm could then be used to shift the relative probabilities of qubit states so that observation of a bitstring corresponding to an optimal or near-optimal solution is highly likely on measurement. This provides a very efficient means of sampling from the near-optimal solutions, one that can potentially surmount the scaling limitations of classical heuristics, since exhaustive representation of the exponentially-scaling search space is possible only on quantum hardware. In previous work using the D-Wave 2000Q quantum annealer, we demonstrated how the protein design problem can be expressed as a combinatorial optimization problem and solved using quantum annealing hardware and hybrid quantum-classical solvers. [11] Critically, we were able to show the approach's applicability to real-world protein design problems without reducing the complexity of the problem.

The D-Wave 2000Q and Advantage systems possess 2,000 and 5,000 qubits and can emulate approximately 64 and 124 fully-connected qubits, respectively [12]. IBM's largest gate-based quantum computer to date allows coupling of any pair of qubits, but has only 65 qubits [13]. Although large problems can be divided into smaller problems that can fit in available qubits using an outer classical algorithm like QBSolv [14], doing so eliminates the quantum scaling advantage. There is therefore considerable interest in developing an intelligent approach to pruning nonproductive rotamers from rotamer optimization problems to allow problems of interest to fit on current and near-future quantum hardware with relatively small numbers of qubits. Such an approach could also help classical rotamer optimization algorithms that are limited by the size of the search space, as well as hybrid algorithms that use an outer classical loop to divide a problem into sub-problems solved by an inner quantum algorithm.

This method used the Rosetta software suite to model these backbone-dependent rotamers and to calculate the one- and two-body interactions between them [8, 15, 16]. Our goal was to find the set of rotamers that minimizes the protein's computed energy, measured in Rosetta Energy Units (REU). Rosetta does this using simulated annealing, in a process called "packing" and "rotamer substitution" [3, 17].

Mapping large protein design problems directly to quantum hardware was limited by a number of factors including noise and the number of qubits available. Even using a hybrid solver proved impractical for large problems as noise and time constraints effectively placed an upper barrier to the size of problems that could be solved. Additionally, we have evidence that the modeling of some atomic interactions, like hydrogen bonds, would be improved with a finer granularity of rotamer sampling, suggesting that our problem has reason to grow even larger [18].

Our goal for this project was to use machine learning to adaptively decrease sample space for arbitrary protein design problems by eliminating rotamers from consideration. Scientists are having rapidly-increasing success using artificial neural networks to design proteins using a variety of representations [19, 20]. We have recently seen success representing proteins by passing contact maps into image-inspired 2D convolutions [21, 22], 3D convolutions on

voxelized representations [23, 24], and even language models on protein sequences [25–27]. However, the representation that interests us the most is the graph-based representation found in graph neural networks [28–30].

Graphs are intuitive representations for protein modeling cases in which the backbone structure is already established, as it is in protein design. In fact, traditional protein modeling tools such as Rosetta use graphs internally to model interactions during their own protocols [8, 31–33]. These residue-centric graphs represent each sequence position as a node, with edges connecting positions that are close in 3D space. Node attributes generally encode the residue's backbone geometry and possibly some representation of its side-chain identity. Edge attributes are used to model the interactions and geometry between residue positions.

Graph neural networks (GNNs) are a class of machine learning models designed to process graph-structured data. While the seminal research on GNNs dates back to the works of Sperduti *et al.* [34], Gori *et al.* [35], and Scarselli *et al.* [36], recent research efforts have led to a rapid growth of the field and have achieved state-of-the-art results on a large variety of applications, ranging from social networks [37–39], to chemistry [40, 41], biology [28, 42, 43], and physics [44].

The growth of the field has led to the development of many diverse GNN architectures, notably including the works in references [45–50]. Of particular interest to this work are those models that can be expressed as message-passing architectures [51]. In particular, message-passing GNNs act on the node attributes of a graph according to the following general scheme:

$$\mathbf{x}'_i = \gamma(\mathbf{x}_i, \square_{j \in \mathcal{N}(i)}\, \phi(\mathbf{x}_i, \mathbf{x}_j, \mathbf{e}_{(j,i)})), \quad \forall i \in \mathcal{V} \tag{1}$$

where $\phi$ is a *message* function that depends on the graph's node and edge attributes (resp. $\mathbf{X}$ and $\mathbf{E}$), $\square$ is any permutation-invariant operation that aggregates messages coming from the neighborhood of $i$, and $\gamma$ is an *update* function (see our Notation section on the next page for the remaining symbols). Intuitively, message-passing GNNs transform the attributes of the graph by exchanging information between neighboring nodes.

While the definition of Eq (1) allows the message function to depend on the edge attribute between a node and its neighbor, the majority of GNN architectures are designed for non-attributed edges. Among those GNNs that are designed to process edge attributes, we mention the Edge-Conditioned Convolutions (ECCs) introduced by Simonovsky and Komodakis [52]. ECCs make use of an auxiliary model called a *filter-generating network* (FGN) that takes as input edge attributes and produces output parameters that replace what conventionally would be the learnable parameters of $\phi$ in Eq (1) that would ordinarily be fixed. ECCs can bring significant advantages when processing graphs for which edge attributes are important and have been used to process molecular graphs [53, 54]. However, the FGN can be difficult to train due to the absence of a strong supervision signal (which is particularly difficult to achieve when stacking many layers) and ECCs are mostly effective in processing edge attributes with a one-hot representation.

In recent years, other types of GNNs have been proposed that process edge attributes directly in the message function, without relying on a FGN. These usually concatenate [55] or sum [56] the edge attributes to the node attributes of the neighbors. In particular, here we consider the work of Xie *et al.* [55], based on concatenation, which we denote as *CrystalConv* in the following.

We note, however, that all of the methods mentioned above suffer from two key issues. First, none of them are designed to take into account the case of symmetric directed graphs with asymmetric edge attributes (*i.e.*, graphs for which the existence of edge $(i, j)$ implies the existence of edge $(j, i)$ and *vice versa*, but the corresponding attributes can differ). This is

particularly relevant for our work due to the geometric nature of our edge attributes: our edges themselves have no directionality but nearly every edge feature has some degree of asymmetry. Second, most existing methods are not designed to update edge attributes, which are considered as static inputs throughout the network. The updating of edge attributes is not a novel idea *per se*, since it was proposed both in the Graph Network model by Battaglia *et al.* [57] and in the Typed Graph Network of Prates *et al.* [58] (where both are works that attempt to unify GNNs in a similar spirit to the message passing framework), but to the best of our knowledge it is seldom applied in practice.

Here we propose XENet, a GNN model that addresses both concerns while also avoid the computational issues introduced by FGNs. XENet is a message-passing GNN that simultaneously accounts for both the incoming and outgoing neighbors of each node, such that a node's representation is based on the messages it receives as well as those it sends. We demonstrate XENet's advantage over ECC and CrystalConv by testing their abilities to eliminate rotamer candidates in real-world protein design problems, with application both to decreasing the solution space that must be searched by classical protein design algorithms, and to decreasing the qubits required for quantum protein design algorithms.

## Materials and methods

**Notation** Let a graph be a tuple $\mathcal{G} = (\mathcal{V}, \mathcal{E})$, with node set $\mathcal{V} = \{1, \ldots, N\}$ and edge set $\mathcal{E} \subseteq \mathcal{V} \times \mathcal{V}$ s.t. $(i, j) \in \mathcal{E}$ is a directed edge from node $i$ to node $j$. Additionally, let $\mathbf{x}_i \in \mathbb{R}^F$ indicate a vector attribute associated with node $i$ and let $\mathbf{e}_{i,j} \in \mathbb{R}^S$ indicate a vector attribute associated with edge $(i, j)$. We indicate the neighborhood of a node with $\mathcal{N}(i) = \{j \mid (j, i) \in \mathcal{E}\}$. Note that in our case we consider symmetric directed graphs, so that the incoming and outgoing neighbors of a node coincide.

To make notation more compact, in the following we denote with $\mathbf{X} \in \mathbb{R}^{N \times F}$ the matrix of node attributes, with $\mathbf{E} \in \mathbb{R}^{N \times N \times S}$ the matrix of edge attributes (we assume the entries of this matrix to be zero if the corresponding edge does not exist), and with $\mathbf{A} \in \{0, 1\}^{N \times N}$ the binary adjacency matrix of the graph.

### XENet

Our architecture, which we refer to as XENet (due to its ability to convolve over both $\mathbf{X}$ and $\mathbf{E}$ tensors), is described by the following Equations:

$$\mathbf{s}_{ij} = \varphi^{(s)}\left(\mathbf{x}_i \| \mathbf{x}_j \| \mathbf{e}_{(i,j)} \| \mathbf{e}_{(j,i)}\right) \tag{2}$$

$$\mathbf{s}_i^{(\text{out})} = \sum_{j \in \mathcal{N}(i)} a^{(\text{out})}(\mathbf{s}_{ij}) \cdot \mathbf{s}_{ij} \tag{3}$$

$$\mathbf{s}_i^{(\text{in})} = \sum_{j \in \mathcal{N}(i)} a^{(\text{in})}(\mathbf{s}_{ij}) \cdot \mathbf{s}_{ji} \tag{4}$$

$$\mathbf{x}_i' = \varphi^{(n)}\left(\mathbf{x}_i \| \mathbf{s}_i^{(\text{out})} \| \mathbf{s}_i^{(\text{in})}\right) \tag{5}$$

$$\mathbf{e}_{(i,j)}' = \varphi^{(e)}\left(\mathbf{s}_{ij}\right) \tag{6}$$

where $\varphi^{(s)}$, $\varphi^{(n)}$, $\varphi^{(e)}$ are multi-layer perceptrons with Parametric Rectified Linear Unit

activations [59], and where $a^{(out)}$ and $a^{(in)}$ are two dense layers with sigmoid activations and a single scalar output.

The core of XENet lies in the computation and aggregation of the *feature stacks* $\mathbf{s}_{ij}$ in Eqs (2)–(4). These are obtained by concatenating the node and edge attributes associated with the incoming and outgoing messages (Eq (2)), so that the multi-layer perceptron $\varphi^{(s)}$ learns to process the two directions separately. The feature stacks are also aggregated separately in the two directions of the flow, using self-attention [60] to compute a weighted sum (Eqs (3) and (4)). The separate representations are concatenated and used to update the node attributes of the graph (Eq (5)). Finally, some additional processing of the feature stacks through $\varphi^{(e)}$ lets us compute new edge attributes that are dependent on the message exchange between nodes (Eq (6)).

## Generating FixbbGCN training data

Here we prepare to apply XENet to a specific protein design problem, as described later in the paper. Our goal is to create a GNN that can analyze a rotamer optimization problem and predict which rotamers are likely to be sampled in the next round of rotamer substitution and which can be omitted. We call this trained network "FixbbGCN".

We used an arbitrary subset of structures from the Top8000 dataset for training [61, 62], which ensures that each protein structure is adequately refined for our use. Our training set used 967 structures (total of 229,776 residue positions) and our validation set used 239 structures (57,584 residue positions). The number of structures we used simply depended on how much CPU time we were willing to commit for generating data.

We ran 5 repeats of the MonomerDesign2019 variant of Rosetta's FastDesign [63, 64] protocol on each structure but only collected training data for the final 4 repeats. We set Rosetta to generate a larger number of more finely-discretized rotamers by passing the '-ex1 -ex2' commandline flags and used Rosetta's REF2015 energy function [16]. This accounts for 16 of the 20 rounds of rotamer substitution, though for this project we only use the data from 4 of the 16 rounds due to score function ramping [63]. We therefore ended up with 919,104 training set elements (229,776 residues x 4 rounds per residue) and 230,336 validation elements.

For this project, rotamers from the 20 amino acids were binned into 54 categories. Alanine and Glycine each had their own bin due to their lack of meaningful $\chi 1$ attributes. Proline was also only assigned one bin despite having two $\chi 1$ rotamer wells [65]. The decision to simplify Proline's $\chi 1$ binning was driven by the high-risk, low-reward action of eliminating Proline rotamers from substitution rounds. Proline is valuable for design but is often represented by relatively few rotamers, so the reward of eliminating Proline is not great. For this reason, we wanted to let our neural network focus its resources on eliminating the other amino acids. The remaining 17 canonical amino acids had three bins each, which correspond to the three $\chi 1$ wells.

For each round of rotamer substitution, we tracked the fraction of time that each rotamer was the representative state for its residue position. At the end of the run, any rotamer bin that held the representative state for more than 0.1% of the run was classified as a 1. All other rotamer bins were classified as a 0. Note that this resulted in a multi-label classification problem where every sample was associated with one or more classes. We also ignored data from the fraction of the simulated annealing trajectories where the simulated temperature was above 1,500 K ($k_B T > 3.0$ kcal/mol).

## FixbbGCN architecture

We refer to this family of networks as FixbbGCN, as the Rosetta rotamer substitution protocol is sometimes called "fixbb". FixbbGCN is schematically represented in Fig 1. The model has

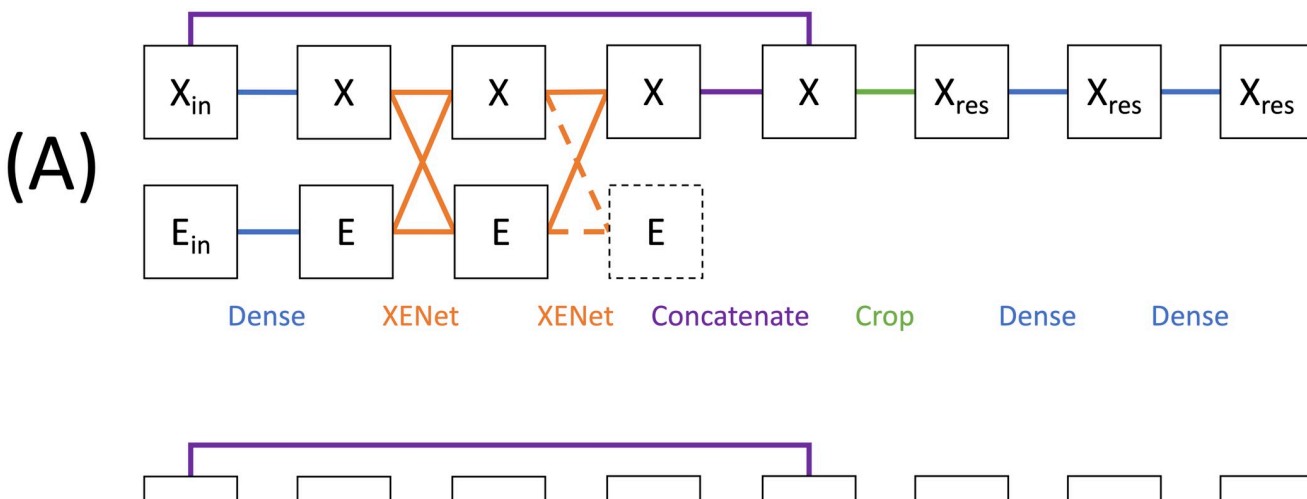

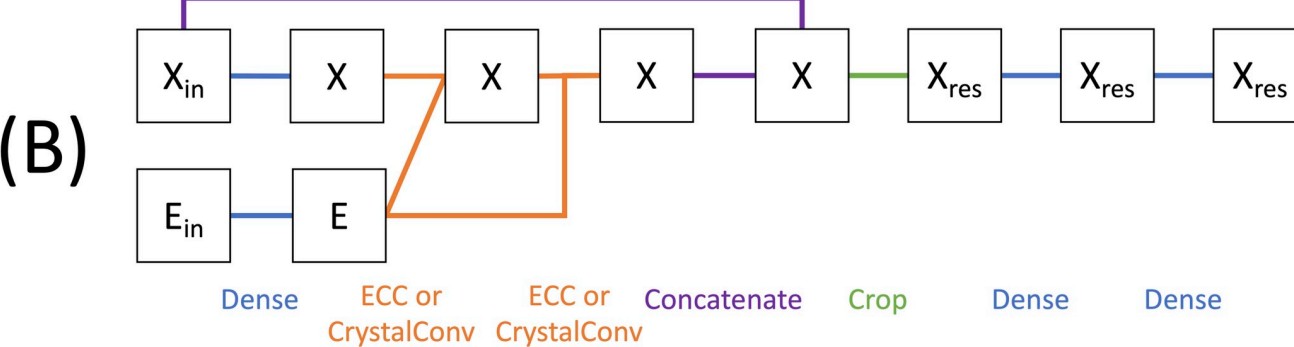

**Fig 1. Schematic representation of FixbbGCNs, the networks used in our experiments.** (A) Example layout for a model with two XENet layers. X denotes node attribute tensor with $X_{in}$ as the input tensor and $X_{res}$ as the single-node subset of the X tensor which represents the protein residue of interest. E denotes the edge attribute tensor with $E_{in}$ as the input tensor. Dotted lines are used to represent operations that are omitted as described in the main text. (B) Example layout for a model with two ECC or CrystalConv layers using the same notation. The A tensor is omitted from this diagram because it never changes.

three input tensors for **X**, **A**, and **E**. The maximum number of nodes per graph representation is $N = 30$, the number of attributes per node is $F = 46$, and the number of attributes per edge is $S = 28$. The output of the model is a 54-dimensional vector which holds one value for each of the rotamer bins described in the "Generating Training Data" section.

For all models, the **X** and **E** tensors are first fed to dense layers. These fully-connected layers only process one node/edge at a time, so that no information flows between nodes or edges. We then apply one or more steps of message passing, using either XENet, CrystalConv, or ECC layers. We used the Spektral package's implementation of the latter two layers. [66].

Fig 1 shows two rounds of message passing but we tested all models with one, two, and three layers (some XENet models were tested up to five layers, as reported in the SI). We note that the output tensor **E** from the final round of XENet is never be used by a future layer. The subset of parameters used to build this final **E** will be implicitly omitted when we tally trainable parameters.

We set FixbbGCN up as a single-node classification problem as opposed to a graph classification problem. Thus, after the message-passing stage, we focus on the unique node that represents the residue of interest being evaluated. We concatenate the output from the final message-passing layer with the original input **X** tensor in an effort to compensate the over-smoothing effect of message passing. We then crop the **X** tensor to only include the node that represents the protein residue of interest. FixbbGCN finishes off by running that single node's data through two more fully-connected layers.

All dense and message-passing layers have ReLU activation functions except for the final dense layer which has a sigmoid activation.

**Hidden layer sizes.**   We benchmarked two XENet candidates as outlined in Table 1. XENet (s) is sized to have the same hidden layer size as the ECC models. XENet (p) is sized to have the same number of trainable parameters as the ECC models. We tuned these parameters by changing $F_h$ and $S_h$, which are the number of channels for the hidden X and E layers, respectively, before the cropping layer. The penultimate dense layer always has 100 channels and the final layer always has 54 channels.

Likewise, we benchmarked two CrystalConv models using the same two normalization techniques and similarly labeled them with (s) and (p). The parameter normalization was not perfect but we got as close as possible without varying hyperparameters between depths of the same type.

Each XENet layer always used two internal stacking layers with $S_h$ channels each. In other words, the $\varphi^{(s)}$ multi-layer perceptrons always had a depth of two.

**Model input.**   FixbbGCN is intended to evaluate a rotamer substitution problem and make predictions about how the attempted solution to the problem will unfold. FixbbGCN does not analyze each rotamer in the set of candidates; instead, it analyzes the current state of the protein model prior to rotamer substitution. This limits the applicability of FixbbGCN to be used only with highly-refined protein models where the states before and after rotamer substitution are expected to be similar. For example, the high-resolution crystal structures used in our benchmarks are expected to maintain their general side-chain behavior even after undergoing rotamer substitution. A counter-example would be *de novo* design on a generated backbone. FixbbGCN could still be used on that project but would require high-resolution model refinement beforehand.

**Node and edge attributes.**   Our input data had 46 node attributes and 28 edge attributes, all of which are listed in S1 Text of the supporting information. Most of these attributes are direct physical characteristics of residues and physical relationships of residue pairs. We also included more advanced analytics in the form of Rosetta score terms.

Many of these attributes require access to the PyRosetta package to compute. [67] These include the Rosetta score terms, hydrogen bond identification, and the residue pair "jump"

**Table 1. Hidden layer sizes and number of trainable parameters for all models.** $F_h$ is the number of channels for hidden X layers and $S_h$ is the number of channels for hidden E layers.

| Convolution | # Layers | Parameters | $F_h$ | $S_h$ |
|---|---|---|---|---|
| ECC | 1 | 100,067 | 49 | 32 |
| ECC | 2 | 181,701 | 49 | 32 |
| ECC | 3 | 263,335 | 49 | 32 |
| CrystalConv (s) | 1 | 31,271 | 49 | 32 |
| CrystalConv (s) | 2 | 44,109 | 49 | 32 |
| CrystalConv (s) | 3 | 56,947 | 49 | 32 |
| CrystalConv (p) | 1 | 109,283 | 125 | 64 |
| CrystalConv (p) | 2 | 188,033 | 125 | 64 |
| CrystalConv (p) | 3 | 266,783 | 125 | 64 |
| XENet (s) | 1 | 30,421 | 49 | 32 |
| XENet (s) | 2 | 47,625 | 49 | 32 |
| XENet (s) | 3 | 64,829 | 49 | 32 |
| XENet (p) | 1 | 92,928 | 128 | 64 |
| XENet (p) | 2 | 179,522 | 128 | 64 |
| XENet (p) | 3 | 266,116 | 128 | 64 |

measurements. A Rosetta "jump" describes the six-dimensional rigid body relationship between the coordinate frames of two protein residues based on their backbone atoms.

Our model was inconsistent in handling angle values. Backbone torsion angles and trRosetta-inspired residue pair angles [68] were represented as single values measured in radians, whereas side-chain chi angles were split into sine and cosine values. This decision was driven by the varying levels of confidence in a neural network's ability to interpret each metric. The trRosetta-inspired measurements were represented as single radian values in the original paper [68], which we felt no need to change. In our experience, neural networks do not benefit from splitting phi/psi angles into sine and cosine, however we do not have deep experience with modeling chi angles. In short, chi angles were split into sine and cosine for this project because we do not yet have confidence that neural networks can efficiently interpret them as single radian values.

**MentenGCN package.** We have created a public Python package in an effort to make protein processing with GNNs more portable and easier to share. MentenGCN [69] has a library of tensor decorators that were used for this project to generate the **X**, **A**, and **E** input tensors directly from Rosetta's protein representation. The configuration class for the GNN used in this paper is available within the MentenGCN package under the name "`Maguire_Grattarola_2021`". Please refer to S1 Text of the supporting information for more detail on how to access this feature.

## Training and evaluating FixbbGCN models

Each model configuration was trained between 6 and 12 times, loosely depending on the amount of resources required to train each model. We show later that the performance of a given architecture generally has narrow variance so we did not see the need to expand this sampling.

Each model was trained using Keras's implementation of the Adam optimizer with a starting learning rate of 0.001 and the binary crossentropy loss function [70, 71]. The learning rate was reduced by a factor of 10 whenever the validation loss plateaued for 2 consecutive epochs (`min_delta = 0.001`). Training was halted whenever the validation loss plateaued for 5 consecutive epochs. We evaluated all models with binary crossentropy and Receiver Operating Characteristic (ROC) area-under-curve (AUC) on our validation set.

## Benchmarking FixbbGCN implementation on classical computer

As we will show in the Results section, the best model observed was XENet (p) with 3 layers. We benchmarked the applicability of this model by using it alongside Rosetta's packing protocol on six backbones of various sizes. For each backbone, we ran each residue position through our model and compared the 54 final values against a tuneable cutoff. Rotamers were eliminated if the final value for their respective bin fell below the cutoff. We performed this benchmark with a range of cutoffs between 0 and 1. We also included a cutoff of -1.0 as a control (so that no rotamers were eliminated, since the sigmoid activation has a minimum of 0). The larger the cutoff, the more aggressively rotamers were eliminated. We ran each cutoff on each structure 10 times and tracked the final Rosetta score in units of Rosetta Energy Units (REU) where more negative is better. Although Rosetta was used for benchmarking purposes, the rational pruning of rotamers offers a benefit to any rotamer optimization method, and could easily be applied to rotamer optimization problems solved with Toulbar2, Osprey, or other software as well.

The Protein Data Bank codes for the six backbones used for this benchmark are 1SFX, 1ECO, 1D4O, 1W2C, 1O4S, and 1PJ5 in order of increasing size. All six of these structures are

also from the top8000 dataset [62] so they are expected to have low homology with the training and validation data used to train the model. Staying consistent with the training data collection, Rosetta built rotamers with the "-ex1 -ex2" commandline flags and used Rosetta's REF2015 score function [16].

## Benchmarking FixbbGCN-XENet implementation on D-Wave advantage quantum annealer

Rational reduction of rotamer candidates by the FixbbGCN can benefit any mapping of design algorithms to quantum computers. As a proof of principle, we used the quantum annealing-based design algorithm called "QPacker", described in Mulligan *et al.* [11]. Briefly, the QPacker approach divides the problem into two steps. First, one- and two-body rotamer energies are pre-computed classically using Rosetta in an $O(N^2D^2)$ operation for $N$ sequence positions and $D$ rotamers per position. The quantum annealer is then used to solve the combinatorial problem, which has an exponentially-scaling $O(D^N)$ solution space, of choosing one rotamer per position so that total energy (the sum of one- and two-body energies for selected rotamers) is minimized. The QPacker approach assigns one qubit per rotamer, and permits rapid and efficient sampling from the bitstrings that provide one-hot encodings of the low-energy rotamer selections. Bitstrings with more than one rotamer selected per position are prohibited by high energetic penalties. We carried out our benchmarks using the D-Wave Advantage 5,000-qubit quantum annealer, using the default 20 $\mu$s annealing schedule. Since the qubits of the D-Wave Advantage are incompletely connected, the 5,000 physical qubits can emulate approximately 124 fully-connected logical qubits. Since our problems involve more than 124 rotamers, we used the QBSolv hybrid algorithm [14], which uses an outer classical loop to set up sub-problems which are solved on the quantum annealer.

As with the classical benchmark, we ran 10 QBSolv attempts (each involving hundreds or thousands of sub-problems on the quantum annealer) for each FixbbGCN cutoff and reported the mean and standard deviation across those 10 attempts. During the classical pre-computation, we also measured Random Access Memory (RAM) usage for each problem size. The RAM usage is expected to scale quadratically with rotamer count due to the need to calculate all residue pair energies between neighboring sequence positions. Since FixbbGCN is able to identify and eliminate nonproductive rotamers prior to the classical energy calculation, this translates to a saving of time and memory in the classical pre-computation in addition to fewer qubits needed for the quantum optimization phase.

This quantum benchmark used all of the same Rosetta parameters and FixbbGCN cutoffs as the classical benchmark. We could not fit the previous test cases on the quantum machine so we used a subset of the smallest problem (protein data bank code: 1SFX). We used Rosetta's LayerSelector tool to design the 10 residues in the core of the protein [72]. All other residue positions were held immutable, decreasing our maximum rotamer count from 63,183 to 5686.

We found that any problem larger than this one would exceed the resources available to it for this benchmark. One relevant drawback of this quantum annealer is that all two-body energies must be exhaustively computed in advance; an $O(R^2)$ operation for a problem with $R$ rotamers. This is another reason the classical benchmarks were able to test larger cases. Rosetta's simulated annealer is able to generate two-body energies on the fly, so the resource usage is linear with the number of considered substitution loops, which in turn is programmed to be linear with the number of rotamers.

## Results and discussion

### FixbbGCN model comparisons

Our goal for this test was to find the graph convolution that would best represent our protein modeling data. XENet is our attempt to engineer a new GNN layer that makes further use of the edge tensors, including updating their features as the result of the convolution. As baseline model for this experiment we considered ECC, since it is one of the first and most widely used GNNs designed to process edge attributes, and we compare it against different configurations of CrystalConv and XENet to ensure a fair comparison. XENet (s) and CrystalConv (s) are normalized by the channel depth of each hidden layer. XENet (p) and CrystalConv (p) are normalized by the trainable parameter count.

The models were tasked with a multi-label classification problem to predict which protein side-chain rotamers would be sampled at a given sequence position during a round of Rosetta's rotamer subsitution protocol with simulated annealing and which could be omitted. [17] We see in Table 2 that the XENet models outperform their ECC and CrystalConv counterparts, although some of the CrystalConv models are in close competition with the best XENet models. In addition to having better loss and AUC scores, XENet convolutions appear to perform better with deeper architectures. XENet slightly improves when the third graph convolution layer is introduced, whereas ECC and CrystalConv exhibit a consistent drop in performance at that depth.

The reasons for these differences in performance can be readily motivated by considering the differences between the models themselves. First, ECC's FGN is an indirect way of processing edge attributes and requires a strong supervision signal in order to be trained effectively, which may not be easy to attain especially within deeper architectures. Second, ECC was often shown to be most effective when processing data with one-hot encoded attributes [52, 53], which is not the case here.

Since CrystalConv does not use a FGN to process the edges, it does not have the same problems as ECC and its performance is more in line with XENet's. However, the asymmetric processing of XENet, paired with its ability to update edge attributes to obtain a richer

**Table 2. Training results.** Mean binary crossentropy loss and mean AUC for trained models. $\sigma$ denotes standard deviation. Lower loss values are considered better whereas higher AUC values are better.

| Convolution | # Layers | Loss | $\sigma$ Loss | AUC | $\sigma$ AUC | # Models |
|---|---|---|---|---|---|---|
| ECC | 1 | 0.188 | 0.004 | 0.9772 | 0.0009 | 8 |
| ECC | 2 | 0.213 | 0.071 | 0.9674 | 0.0224 | 6 |
| ECC | 3 | 5.442 | 0.462 | 0.6248 | 0.0284 | 6 |
| CrystalConv (s) | 1 | 0.173 | 0.001 | 0.9807 | 0.0003 | 8 |
| CrystalConv (s) | 2 | 0.155 | 0.002 | 0.9844 | 0.0003 | 8 |
| CrystalConv (s) | 3 | 4.520 | 0.258 | 0.6872 | 0.0179 | 8 |
| CrystalConv (p) | 1 | 0.158 | 0.002 | 0.9837 | 0.0005 | 8 |
| CrystalConv (p) | 2 | 0.145 | 0.002 | 0.9865 | 0.0004 | 8 |
| CrystalConv (p) | 3 | 5.522 | 0.345 | 0.6238 | 0.0368 | 8 |
| XENet (s) | 1 | 0.155 | 0.001 | 0.9844 | 0.0003 | 10 |
| XENet (s) | 2 | 0.147 | 0.002 | 0.9860 | 0.0005 | 8 |
| XENet (s) | 3 | 0.143 | 0.001 | 0.9868 | 0.0002 | 8 |
| XENet (p) | 1 | 0.143 | 0.001 | 0.9869 | 0.0002 | 10 |
| XENet (p) | 2 | 0.137 | 0.002 | 0.9878 | 0.0004 | 12 |
| XENet (p) | 3 | 0.134 | 0.002 | 0.9883 | 0.0004 | 8 |

representation, make it more suitable for this particular type of data and results in a better overall performance in all configurations.

We show in Fig 2 that XENet can even handle depths of 4 and 5 GNN layers. The additional layers did not give us an advantage in validation loss; however, deeper architectures will theoretically be more advantageous for use cases that require more expansive message passing than our benchmark. For this reason, the mere ability to handle deeper architectures may prove to be a strength of XENet. XENet did encounter occasional failures with the deeper architectures but the majority of deeper models finished with competitive validation losses. We did not test CrystalConv or ECC with architectures of 4 or 5 layers due to their lack of success with 3 layers.

## Quantum FixbbGCN benchmark

With trained models in hand, we wanted to see how much they can decrease the classical and quantum resource costs for our quantum annealing use cases. We wrapped the best model for each architecture in Rosetta rotamer-elimination machinery and named it FixbbGCN ("fixbb" is a popular name for Rosetta's fixed-backbone packing protocol).

We cannot run full-sized quantum benchmarks for the same reason that this project was motivated: our protein design benchmarks are too large to be run on the quantum computers. The best we can currently do is use FixbbGCN to design a subset of the protein on the quantum annealer and save the larger problems for the classical benchmark presented later in the article.

For this test, we needed a very small problem size. We took the smallest test case from our benchmark set but restricted sampling to only include the core of the protein. We used Rosetta's definition of the core of the protein, which identified 10 residue positions that were sufficiently isolated from solvent exposure.

We chose this benchmark because the core is the most combinatorially challenging part of the protein to design. Rosetta samples core rotamers more finely than solvent-exposed residues so the rotamer count per position is higher. Additionally, these residue positions tend to have more neighbors, resulting in a more complex energy optimization problem.

XENet shows in Fig 3 an ability to decrease the rotamer count to roughly 60% before the dip in Rosetta score appears. ECC drops in quality near 70% and CrystalConv drops near 64%.

We did not report runtime for this benchmark because we had no way to decouple time spent running the annealer from time spent sending our data over the internet and waiting in the quantum computer's queue. However, we expect that setup time will correlate linearly with RAM usage as both have quadratic relationships with the rotamer count. Quantum annealing time itself is currently limited by the coherence time of the D-Wave's superconducting qubits, but represents a small minority of the total computing time. However, the benefit for qubit usage can be quantified: given the QPacker algorithm, which uses $ND$ logical qubits to represent a problem with $N$ variable amino acid positions and $D$ rotamers per position, the fold reduction in rotamer count is equal to the fold reduction in the number of logical qubits needed (i.e. if FixbbGCN is used to reduce the number of rotamers by 40%, it reduces the number of logical qubits by 40% as well).

Using RAM as our guide, XENet is able to reduce our problem's memory consumption to 32% before the decrease in design quality appears. The CrystalConv model came close with a decrease to 36% memory consumption and the ECC only model shrank the problem to 43% memory consumption.

Finally the reduction in the size of the solution space can also be quantified. Given that there are $D^N$ possible solutions to a packing problem of $N$ positions and $D$ rotamers per

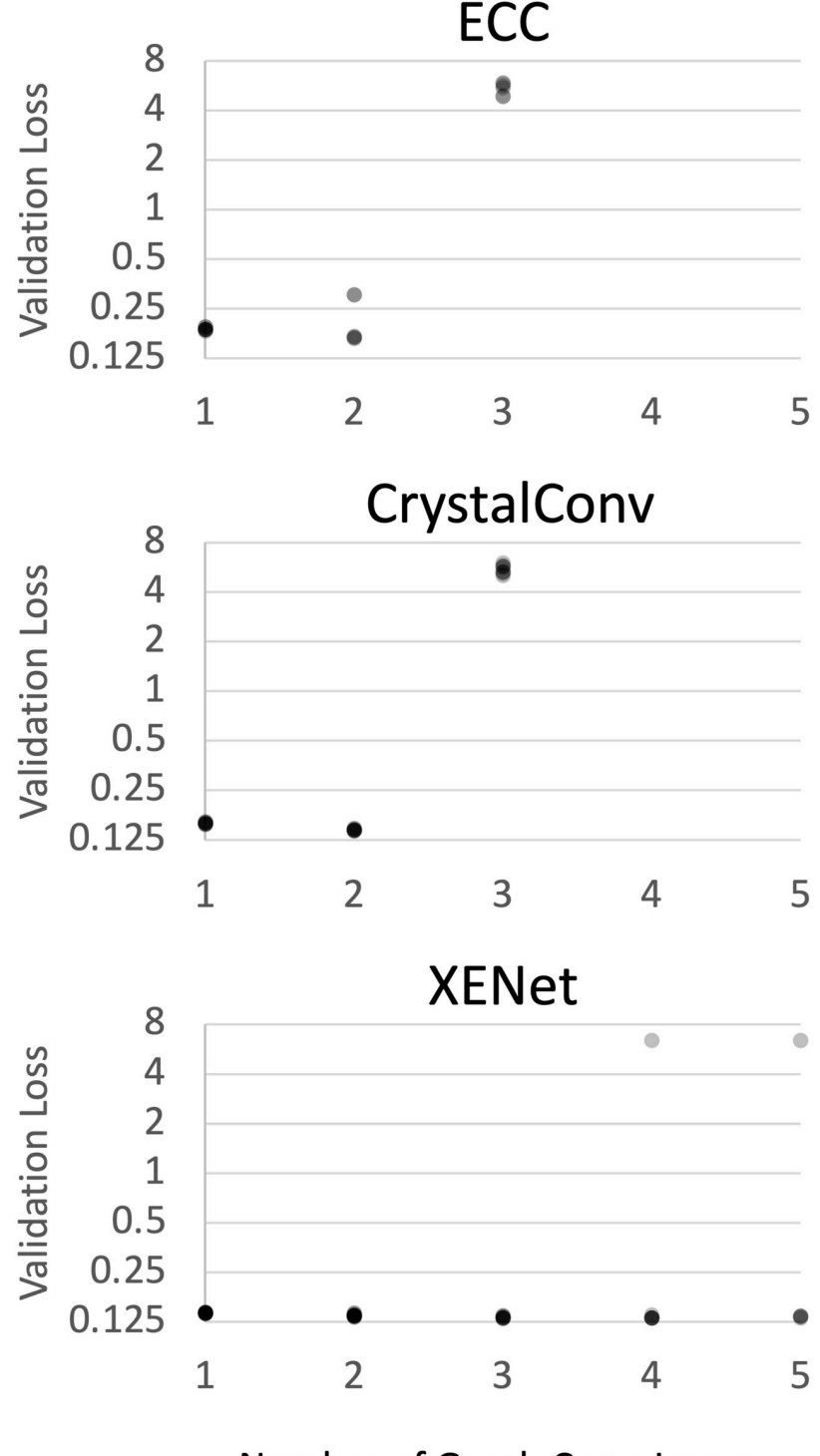

**Fig 2. Depth comparison with fixed parameter count.** We plot the losses of all trained ECC, CrystalConv (p), and XENet (p) models against the number of graph convolutional layers in each model. Transparency was applied to the points to help illustrate density. ECC and CrystalConv have no points with 4 or 5 layers.

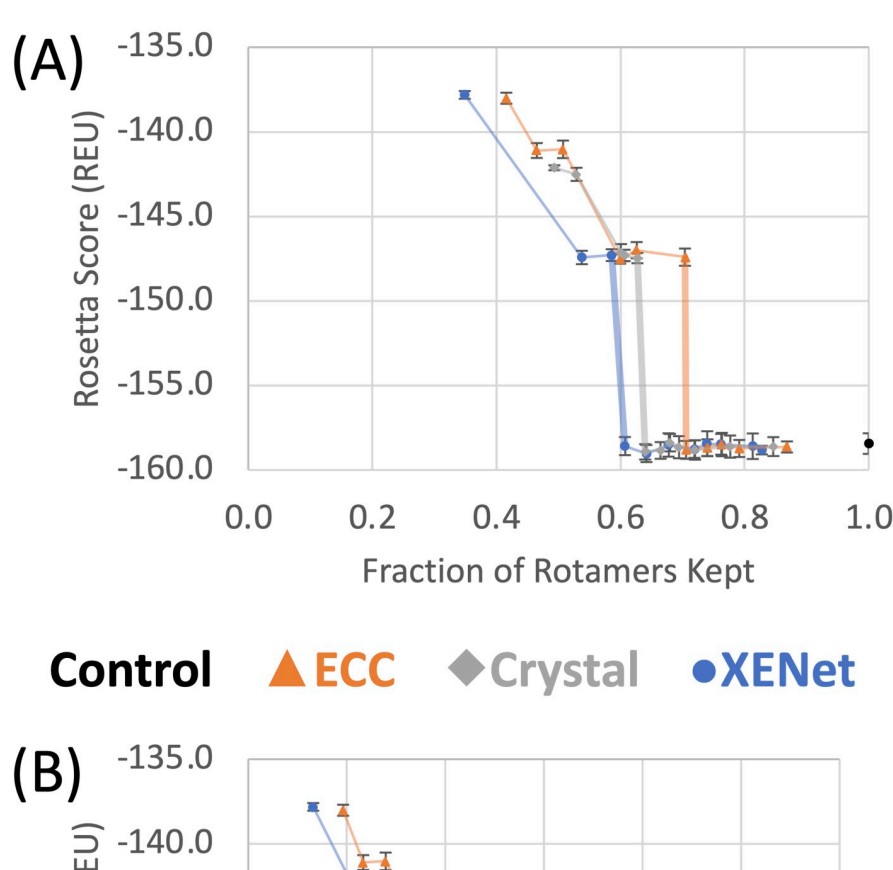

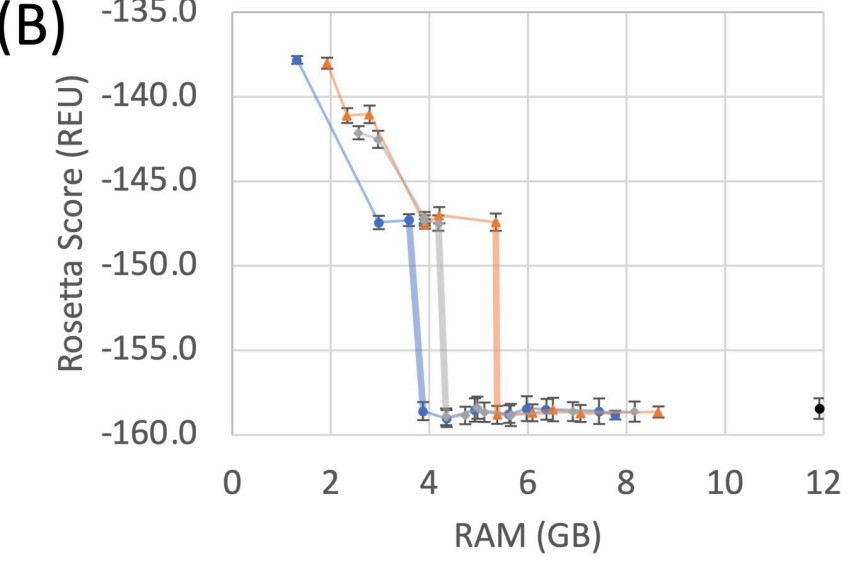

**Fig 3. Quantum FixbbGCN benchmark results.** (A) Mean Rosetta Scores for various cutoffs and convolutions types. Lines connect points of the same convolution and the line to the first drop in design quality is drawn thick. X-axis values are the number of surviving rotamers for a cutoff/convolution pair divided by the number of rotamers in the control case. (B) Same results as (A) but plotting against annealer memory usage instead of rotamer count. Both y-axes are truncated for the sake of readability.

position, a 40% reduction in the number of rotamers per position means that the total problem space is reduced to $0.6^N$ of what it was. In the case of this quantum benchmark, with $N = 10$ positions, the FixbbGCN network could scale the solution space by a factor of 0.006 (i.e. to about 1/165th its original size). Although this translates to a linear decrease in quantum resource usage, for classical algorithms this represents a massive reduction in the amount of

solution space that must be searched and the amount of computing power needed to find the global optimum, as discussed in the next section.

## Classical FixbbGCN-XENet benchmark

The goal for the final benchmark was to assess to what extent XENet's pattern observed in the quantum benchmark persists for full-sized use cases. Unfortunately, these full-sized design cases are too large for us to run on quantum computers so we ran these benchmarks using Rosetta's simulated annealer. This is the best we can do with current technology but hopefully a more complete test will be possible someday.

Similar to the quantum benchmark, this benchmark applies the XENet classifier with various cutoffs to Rosetta's set of rotamers for six different protein design problems. This time, however, the entire protein structures are being designed. Rotamers are pruned if their predicted value from the classifier is below the cutoff. The "control" data point with the largest rotamer count for a given use case is the standard Rosetta packing protocol with no influence from the classifier.

We see in Fig 4 that we can use FixbbGCN to decrease the number of rotamers without a loss in design quality to a limited extent. The Rosetta score will generally stay in range of the control data down to the range of 55–60% of the original rotamer count.

Fig 4 illustrates how larger problems generally behave more predictably than smaller problems. For the control (x-axis value = 1.0), the magnitude of the error is roughly 0.6% for the smallest problem and roughly 0.2% for the largest problem. Additionally, the impact of FixbbGCN appears to follow a more predictable pattern for larger sizes. One interpretation of this increasing predictability is that the larger proteins have more buried positions (unexposed

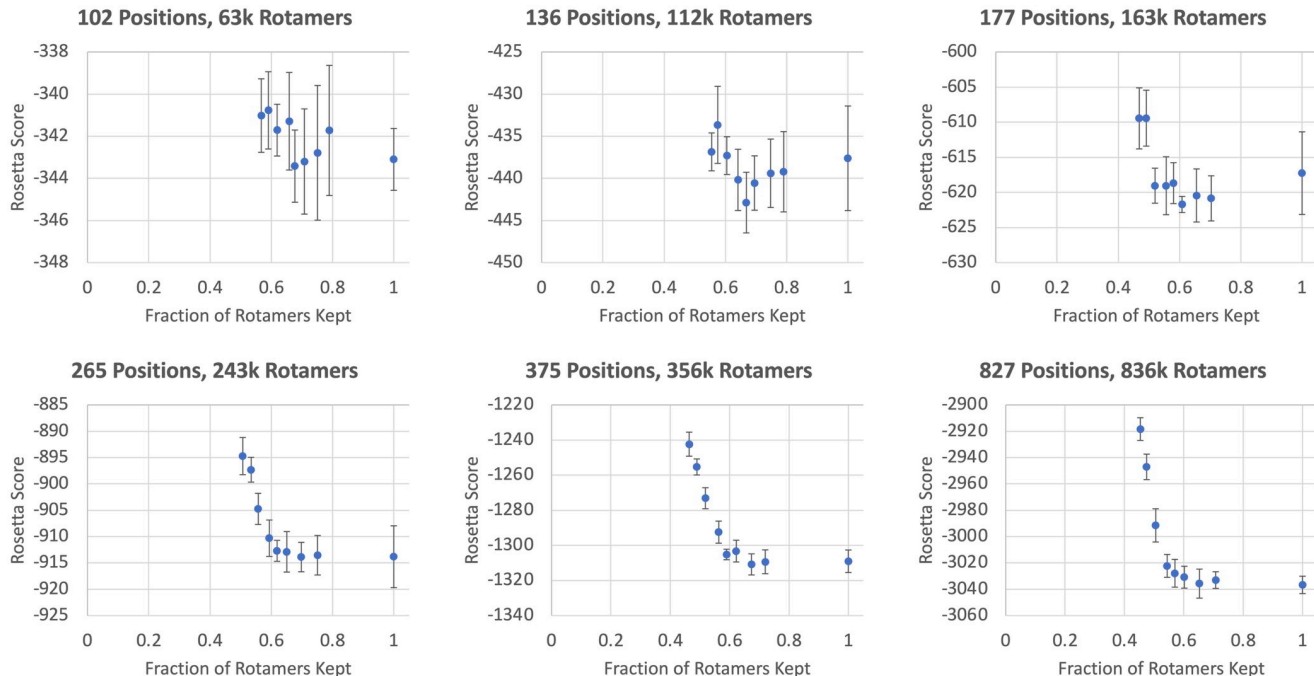

**Fig 4. FixbbGCN benchmark results.** Results of running Rosetta's rotamer substitution protocol on six different protein backbones. FixbbGCN was used with various cutoffs to decrease the total rotamer count of each sample. The mean Rosetta scores (measured in REU) and standard deviations are displayed for each cutoff. The y-axes are truncated for the sake of readability. Protein data bank codes from left to right are (top row) 1SFX, 1ECO, 1D4O and (bottom row) 1W2C, 1O4S, 1PJ5.

to solvent). Buried positions are traditionally considered to be challenging to sample due to their inherently large number of neighbors. While this higher difficulty may exist, there may also be a higher degree of sampling predictability due to the confined local conformation of the buried pocket. We do not have enough evidence to do more than speculate at this time; future research would be required to dive into the intricacies of the rotamer substitution behavior.

The results in Fig 4 supports the idea that FixbbGCN's ability to eliminate rotamers for small problem sizes translates to larger problem sizes too. It is up to the user to decide how risky they want to be with FixbbGCN, but our results suggests that decreasing rotamer counts to roughly 60% is safe. As discussed in the previous section, for a problem with $N$ variable positions, this represents a shrinkage of the size of the solution space by a factor of $0.6^N$, exponentially decreasing the necessary length of trajectories searching the solution space to find low-energy solutions. For the examples shown in Fig 4, $N$ ranges from 102 to 827 sequence positions, meaning that FixbbGCN was able to reduce the size of the solution space by a factor of $4.3 \times 10^{23}$ to $2.9 \times 10^{183}$, massively improving the probability of finding low-energy states in a finite-length Monte Carlo trajectory.

## Conclusion

Graph neural networks have great potential for modeling residue-level protein interactions. We show that our new convolution, XENet, can model residue-level environments better than existing methods ECC and CrystalConv. Not only does the usage of XENet result in lower validation losses, but we show that XENet can withstand deeper architectures.

To demonstrate XENet's value, we have used it to create a tool (FixbbGCN) capable of fitting larger protein design problems onto quantum computers by eliminating side-chain conformations that are unlikely to be selected by a rotamer optimization algorithm. XENet was consistently able to reduce rotamer counts by 40% without loss in design quality. As a result, we measured a 68% decrease in total problem memory consumption, which has a quadratic relationship with rotamer count. Most importantly, we were able to achieve a massive reduction in the size of the solution space, from 2 orders of magnitude for a small problem with 10 variable positions to 183 orders of magnitude for a large problem with 827 variable positions. This ability to shrink the solution space exponentially reduces qubit consumption linearly on quantum computing hardware, making it feasible to solve more difficult protein design problems on more modest, nearer-term quantum computers, and facilitating future research to determine the ultimate utility of quantum computers to protein design. The exponential reduction in the size of the solution space also results in an exponential reduction in the number of solutions that must be searched using classical heuristic methods such as those implemented in Rosetta in order to find near-optimal solutions, as well as in the computing power needed to find the global optimum using exact solvers like those implemented in Toulbar2 or Osprey. More broadly, we anticipate that the XENet convolutional architecture will be widely applicable to many problems in protein modeling.

## Supporting information

**S1 Table. Quantum benchmark results: XENet.** Results of Core Redesign on the quantum computer with CrystalConv. $\sigma$ denotes standard deviation.
(CSV)

**S2 Table. Quantum benchmark results: ECC.** Results of Core Redesign on the quantum computer with CrystalConv. $\sigma$ denotes standard deviation.
(CSV)

**S3 Table. Quantum benchmark results: CrystalConv.** Results of Core Redesign on the quantum computer with CrystalConv. $\sigma$ denotes standard deviation.
(CSV)

**S4 Table. Training loss raw data.** The data points for the means and standard deviations reported in Table 2 (columns 3 and 4) of the main text.
(CSV)

**S5 Table. AUC raw data.** The data points for the means and standard deviations reported in Table 2 (columns 5 and 6) of the main text.
(CSV)

**S6 Table. Quantum benchmark raw data.** These are the Rosetta energies used to comprise the averages in the main text. Note the -1 cutoff value from XENet table was used as the control for the entire experiment. No rotamers were eliminated with that cutoff so it is a global control.
(CSV)

**S1 Text. Node and edge attributes for FixbbGCN.**
(TXT)

**S2 Text. Data availability.** This section attempts to comply with PLOS Computational Biology's data policy. In addition to the downloadables mentioned here, we provide individual data points in upcoming tables.
(TXT)

## Acknowledgments

We thank Dr. Andrew Leaver-Fay and Brian Coventry for various Rosetta developments that made our workflow easier and Sergey Lyskov and Dan Farrell for their assistance in overcoming technical hurdles. We also thank D-Wave Systems, Inc. for useful discussions and support.

## Author Contributions

**Conceptualization:** Jack B. Maguire.

**Data curation:** Jack B. Maguire.

**Formal analysis:** Jack B. Maguire.

**Investigation:** Jack B. Maguire.

**Methodology:** Jack B. Maguire.

**Project administration:** Jack B. Maguire.

**Resources:** Jack B. Maguire.

**Software:** Jack B. Maguire, Vikram Khipple Mulligan, Eugene Klyshko, Hans Melo.

**Supervision:** Jack B. Maguire.

**Validation:** Jack B. Maguire.

**Visualization:** Jack B. Maguire.

**Writing – original draft:** Jack B. Maguire, Daniele Grattarola.

**Writing – review & editing:** Jack B. Maguire, Daniele Grattarola, Vikram Khipple Mulligan, Eugene Klyshko, Hans Melo.

**funding-acquisition:** Hans Melo.

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
