## [Decision Letter · Decision Letter 0]

1 Jul 2021

Dear Dr. Maguire,

Thank you very much for submitting your manuscript "XENet: Using a new graph convolution to accelerate the timeline for protein design on quantum computers" for consideration at PLOS Computational Biology.

As with all papers reviewed by the journal, your manuscript was reviewed by members of the editorial board and by several independent reviewers. In light of the reviews (below this email), we would like to invite the resubmission of a significantly-revised version that takes into account the reviewers' comments.

We cannot make any decision about publication until we have seen the revised manuscript and your response to the reviewers' comments. Your revised manuscript is also likely to be sent to reviewers for further evaluation.

We apologize for the delay.

With respect to the comments from reviewer 1, we recognize that some of the comments may be beyond the scope of your manuscript. However, please do your best to address their concerns.

Sincerely,

Joanna Slusky, Ph.D.

Guest Editor

PLOS Computational Biology

Nir Ben-Tal

Deputy Editor

PLOS Computational Biology

Reviewer's Responses to Questions

**Comments to the Authors:**

Reviewer #1: This is a very interesting paper. Unfortunately, the authors do not describe the quantum computing hardware, nor how the benchmarks were run on a quantum computer. What was this hardware, how many qubits, how much ECC, what QC architecture? How was noise handled? What about decoherence? The paper does not contain enough details. Second, running times could not be compared in the experimental designs. Third, the reductions in problem size, while interesting, are quite modest. Fourth, the Rosetta paradigm is notoriously difficult to parallelize beyond naive/embarrassing parallelism. It does not even use GPUs. A thorough discussion of other (more parallelizable) algorithmic paradigms for protein design (such as Hallen et al.

J Comput Chem. 2018 Nov 15;39(30):2494-2507) is warranted.

Tom Schiex has showed and published that as the size of sequence increases, Rosetta almost never finds the GMEC, and that the number of sequences between the GMEC and the Rosetta answer is often in the tens of millions. QC could help with this if it is faster and more accurate. Can you show that QC addresses these flaws, improving not only speed, but accuracy?

The authors have an exciting idea, and when the issues above have been addressed it could be submitted to a journal. In its current form, the MS is too far from the standard of engineering, computer science, or protein design to be formally published.

Reviewer #2: This is an interesting utilization of graph neural networks to the problem of protein design with a view toward performing design on quantum computers. The description of the methods are well written and the direct comparison to other forms of graph networks are an important feature of the paper.

I have only minor questions/comments which the authors could address:

1) it's odd to use the dihedral angles for the node and edge features in radians for phi,psi, CA1-CB1-CB2-CA2, etc but use sin and cos for the side-chain dihedral angles. It may not make much difference but what would happen if sin and cos were used for all the dihedral angles?

2) It is not quite true that different proteins in the top8000 set are distantly related to each other. In fact there are 1531 proteins in top8000 that have a homologue with greater than 50% sequence identity in top8000. Of the proteins used in this study, 1EDO has a 54% homologue in top8000 (3OSUB). Chain 1O4SA has a 40% homologue. This probably does not affect your results but you should be aware of it. You can use our PISCES server to figure this out: http://dunbrack3.fccc.edu/pisces/PISCES.php, and select the 2nd option where you can upload a list of chains (in the format 1ABCA) and select the option to print out the percent ids of the returned list. Or use these links where i used 99% to cull the list (which removed some 100% identical chains - see the log file). A few chains are in obsolete PDB entries. The percent identity file has the percent identities among all homologous pairs in top8000 (as detected by hhpred).

Download list of PDB chains:

http://dunbrack3.fccc.edu/pisces/users/roland.dunbrack@gmail.com/cullpdb_pc99.0_res0.0-5.0_len40-10000_R0.5_Xray+Nmr+EM_d2021_06_02_chains7886

Download FASTA sequence file:

http://dunbrack3.fccc.edu/pisces/users/roland.dunbrack@gmail.com/cullpdb_pc99.0_res0.0-5.0_len40-10000_R0.5_Xray+Nmr+EM_d2021_06_02_chains7886.fasta

Download sequence identities:

http://dunbrack3.fccc.edu/pisces/users/roland.dunbrack@gmail.com/cullpdb_pc99.0_res0.0-5.0_len40-10000_R0.5_Xray+Nmr+EM_d2021_06_02_chains7886.pc

Download input PDB list:

http://dunbrack3.fccc.edu/pisces/users/roland.dunbrack@gmail.com/uploaded107

Log file:

http://dunbrack3.fccc.edu/pisces/users/roland.dunbrack@gmail.com/cullpdb_pc99.0_res0.0-5.0_len40-10000_R0.5_Xray+Nmr+EM_d2021_06_02_chains7886.log

3) Proline does have two rotamers (chi1 = +30; chi1=-30). Why ignore that?

4) In Figure 4, the larger chains behave more predictably (a fall in energy at 60% rotamer coverage) but the smaller chains do not. Can you comment? (also list the PDB codes for the plots in the caption or above each plot).

5) The node and edge features depend on the rotamer at each node and each pair of nodes. For instance, the side-chain/backbone or side-chain/side-chain hydrogen bonds must depend on the rotamer at each position (including chi2,chi3,chi4). So is the program run once for each rotamer of each residue type at each position so its features are determined? Then what are the values of the edge features, some of which depend on the rotamers/residue types at the neighboring nodes (up to N=30). Perhaps a diagram of the features and how they relate to the output (probability of a rotamer being included in a Rosetta design run) would be helpful.

6) Very minor - I know the rotamer library is sort of part of the furniture in Rosetta :-) but it is a separate thing and Max's paper [Shapovalov and Dunbrack, 2011] can be cited when the rotamers are heavily discussed in a paper.

Reviewer #3: Summary: XENet presents an improved architecture for GNN-based networks informed by message-passing operations. The graph convolution contributions incorporate potentially asymmetric edge attributes representing physical or chemical features. XENet is applied as a preprocessing optimisation for a rotamer substitution protocol and then compared to existing CrystalConv and ECC architectures. In both classical and quantum computing benchmarks, XENet can reduce the problem size by a large factor before seeing a significant loss in data quality.

Comments: The submission clearly defines its scope of current application to rotamer sample reduction and acknowledges previously established literature. Initial definitions are coherent, and data, figures and benchmarks are accessible to a non-specialist audience. XENet's contributions to message-passing GNNs and applications to optimising rotamer substitution are generally well-evidenced with potential improvements (see below).

Specific concerns remain to be revised:

- The motivations for using an quantum annealer appear inadequately expressed in the text. While the paper acknowledges previous work by both the authors and others concerning quantum computing, an additional prefacing paragraph asserting the usefulness and advantages of near-term quantum computing may improve the overall contextualisation of the benchmarks and conclusions.

- The cause of resource limitations for the fixbbGCN benchmarks could be expressed further. XENet is introduced as a suitable contribution to allow more significant protein design problems onto near-term quantum computers. However, the clarification within the quantum fixbbGCN benchmarks uses the smallest test case from the benchmark set. Is the limitation for the usable test case determined by any previous trials or estimations? If possible, perhaps a clarification on the differing tractability between quantum and classical approaches may help. Additionally, would it be feasible to estimate resource usage, perhaps by memory, for larger test cases within the benchmark set?

Other minor concerns for consideration:

- Definitions of (s) and (p) marking the same number of hidden layers and trainable parameters could be rephrased or moved, for the sake of clarity, to include CrystalConv as used in both Tables 1 and 2.

- Sparse grammatical errors are present in the author summary ("graphs data structures are ubiquitous [...]") and "Quantum FixbbGCN Benchmark" subsection ("[...] decrease the sizes our quantum annealing use cases.").

**Have the authors made all data and (if applicable) computational code underlying the findings in their manuscript fully available?**

Reviewer #1: **No: **See review

Reviewer #2: Yes

Reviewer #3: Yes

PLOS authors have the option to publish the peer review history of their article (what does this mean?). If published, this will include your full peer review and any attached files.

Reviewer #1: No

Reviewer #2: **Yes: **Roland Dunbrack

Reviewer #3: **Yes: **Samuel Lim
---

## [Decision Letter · Decision Letter 1]

14 Sep 2021

Dear Dr. Maguire,

We are pleased to inform you that your manuscript 'XENet: Using a new graph convolution to accelerate the timeline for protein design on quantum computers' has been provisionally accepted for publication in PLOS Computational Biology.

Best regards,

Joanna Slusky, Ph.D.

Guest Editor

PLOS Computational Biology

Nir Ben-Tal

Deputy Editor

PLOS Computational Biology

Reviewer's Responses to Questions

**Comments to the Authors:**

Reviewer #2: The authors have adequately responded to my previous comments.

Reviewer #3: The authors have adequately addressed all of my concerns.

**Have the authors made all data and (if applicable) computational code underlying the findings in their manuscript fully available?**

Reviewer #2: Yes

Reviewer #3: Yes

PLOS authors have the option to publish the peer review history of their article (what does this mean?). If published, this will include your full peer review and any attached files.

Reviewer #2: **Yes: **Roland Dunbrack

Reviewer #3: No

---

## [Editor Report · Acceptance letter]

23 Sep 2021

PCOMPBIOL-D-21-00765R1 

XENet: Using a new graph convolution to accelerate the timeline for protein design on quantum computers

Dear Dr Maguire,

I am pleased to inform you that your manuscript has been formally accepted for publication in PLOS Computational Biology. Your manuscript is now with our production department and you will be notified of the publication date in due course.

With kind regards,

Zsofi Zombor
